# EXPLORING TARGET REPRESENTATIONS FOR MASKED AUTOENCODERS

**Xingbin Liu**[1,2*]   **Jinghao Zhou**[2*]   **Tao Kong**[2*]   **Xianming Lin**[1†]   **Rongrong Ji**[1]

[1]Xiamen University    [2]ByteDance

## ABSTRACT

Masked autoencoders have become popular training paradigms for self-supervised visual representation learning. These models randomly mask a portion of the input and reconstruct the masked portion according to assigned target representations. In this paper, we show that a careful choice of the target representation is unnecessary for learning good visual representation. Driven by this observation, we propose a multi-stage masked distillation pipeline and use a randomly initialized model as the teacher, enabling us to effectively train high-capacity models without any effort to carefully design the target representation. On various downstream tasks of classification, transfer learning, object detection, and semantic segmentation, the proposed method to perform masked knowledge **d**istillation with **bo**otstrapped **t**eachers (**dBOT**) outperforms previous self-supervised methods by nontrivial margins. We hope our findings, as well as the proposed method, could motivate people to rethink the roles of target representations in pre-training masked autoencoders. The code and pre-trained models are publicly available at https://github.com/liuxingbin/dbot.

## 1 INTRODUCTION

**M**asked **I**mage **M**odeling (MIM) (He et al., 2022; Wei et al., 2022a; Baevski et al., 2022; Zhou et al., 2021) has recently become an active research topic in the field of visual representation learning and establishes strong performance for vision recognition tasks, *e.g.*, image classification, object detection, and semantic segmentation, which also surpasses traditional supervised learning (Touvron et al., 2021) mechanism. To be specific, MIM randomly masks a portion of the input and then reconstructs the masked portion according to the transformed target, formulated as

$$\min_{\theta} \ \mathbb{E}_{x \sim \mathcal{D}} \ \mathcal{M}(\mathcal{T}(x \odot (1 - M)), f_{\theta}(x \odot M)), \tag{1}$$

where "$\odot$" means element-wise product; $M$ is the *patch mask*; "$x \odot M$" represents "unmasked patches" and vice versa; $f_{\theta}(\cdot)$ is the learnable network to be pre-trained; $\mathcal{T}$ is the transformation function generating the reconstructed target. $\mathcal{T}$ can either be a parameterized network or a traditional image feature transformation method; $\mathcal{M}(\cdot, \cdot)$ is the similarity measurement, *e.g.*, $l2$-distance (He et al., 2022). A masked image passed through the network $f_{\theta}(x \odot M)$ to reconstruct the visual representation of the intact image with transformation $\mathcal{T}(x \odot (1 - M))$.

A crucial problem of MIM is how to choose the reconstructed target, *i.e.*, $\mathcal{T}(\cdot)$ in Eq. (1). Previous methods use disparate teacher networks to generate the reconstruction target. BEiT (Bao et al., 2022) employs a pre-trained DALL-E (Ramesh et al., 2021) as the teacher network. In MaskFeat (Wei et al., 2022a), authors use HOG (Dalal & Triggs, 2005), MoCo (He et al., 2020) and DINO (Caron et al., 2021) features to perform MIM; MVP (Wei et al., 2022b) employs a multi-modality model, CLIP (Radford et al., 2021), which is pre-trained by rich image-text pairs. MAE (He et al., 2022) uses image pixels as the target, which functions likewise to a randomly initialized teacher network, as demonstrated in Appendix B.1. iBOT (Zhou et al., 2021) and data2vec (Baevski et al., 2022) use the exponential moving average (EMA) strategy to update teacher's parameters $\phi$. Though different methods differ in their architectural designs and optimization, the choice of the teacher network lies crucial for each method and calls for a systematic study. In this work, we paraphrase a term **M**asked

---

*Equal contribution. †Corresponding author(linxm@xmu.edu.cn).

**K**nowledge **D**istillation (**MKD**) to focus our discussion on a special case of MIM where the target is generated by a parameterized network (teacher network), *i.e.*, $\mathcal{T}(\cdot) = h_\phi(\cdot)$. In this setting, $\mathbf{T}$ is the teacher network, and $f$ is the student network.

The purpose of our work is to investigate *whether a careful design of the teacher network for MKD matters*. Such exploration is nontrivial given that different teacher networks contain different knowledge we endued into the teacher network, which may induce diverse behaviors for the student networks. And the painstaking selection of the target representations in the field of MIM. To this end, we compare student networks distilled by four teacher networks with different computation pipelines, *i.e.*, DINO (Caron et al., 2021) for contrastive learning, MAE (He et al., 2022) for masked autoencoding, DeiT (Touvron et al., 2021) for supervised learning, and DALL-E (Ramesh et al., 2021) for autoregressive generation. Four teachers are all pre-trained on ImageNet-1K for a fair comparison. To our surprise, although the behaviors of the teacher networks are very different, the distilled student networks share similar characters after several stages of MKD: **(i)** the performance variance between student networks distilled from different teachers rapidly decreases. **(ii)** the model weights and output features across layers within the networks share similar properties.

Such observations indicate that the design of target representation is not essential when pre-trained with multi-stage, *i.e.*, *teacher networks do not matter with multi-stage masked knowledge distillation*. Exceptionally, we use a randomly initialized model as teacher to perform multi-stage masked knowledge distillation, and find that it performs as well as those initialized by pre-trained models with the exact same settings! Using a *random model* as teachers not only avoids an extra pre-training stage, but also alleviates the painstaking selection of the target representations.

Based on the above studies and observations, we naturally propose to perform masked knowledge **d**istillation with **bo**otstrapped **t**eachers, short as **dBOT** 🤖. Specifically, masked knowledge distillation is performed repeatedly in multiple stages. At the end of each stage, we assign the student's weight to the teacher and re-initialize the student's weight to continue masked knowledge distillation. With simple yet effective design that enables pre-training starting from randomly initialized teachers, dBOT achieves 84.5%, 86.6%, and **88.0**% top-1 fine-tuning accuracy on ImageNet-1K (Deng et al., 2009) with ViT-B/16, ViT-L/16, and ViT-H/14, respectively, significantly surpassing previous states of the art, MAE. Beyond that, dBOT achieves 52.7 and **56.0** AP$^{\text{box}}$ for object detection on COCO (Lin et al., 2014), as well as 49.5 and **54.5** mIoU for semantic segmentation on ADE20K (Zhou et al., 2017), with ViT-B/16 and ViT-L/16 respectively. We also explore MKD with teachers of larger sizes, further boosting model performances on various visual tasks.

## 2 RELATED WORK

### 2.1 SELF-SUPERVISED VISUAL LEARNING

Self-supervised learning is an active research topic recently. Early practices revolve around contrastive learning (He et al., 2020; Chen et al., 2020; Grill et al., 2020; Caron et al., 2020; 2021) where the model output features of images transformed by different data augmentations are pulled together. With the development of Masked Language Modeling (MLM) in language pre-training (Devlin et al., 2019), researchers also introduce the training strategy of masked reconstruction to visual pre-training. BEiT (Bao et al., 2022) uses the DALL-E (Ramesh et al., 2021) to encode an image patch as the target for model reconstruction. iBOT (Zhou et al., 2021) uses an online teacher shifting the target from offline to online to make the target semantic meaningful. In addition to using the token obtained from offline or online model as reconstruct target, MAE (He et al., 2022), SimMIM (Xie et al., 2022), and MaskFeat (Wei et al., 2022a) achieve good performance in masked-image reconstruction using low-level pixels or HOG (Dalal & Triggs, 2005) features. Among them, MAE uses an asymmetric encoder-decoder structure greatly increasing the training efficiency. data2vec (Baevski et al., 2022) demonstrates good generalizations on three modalities (vision, speech, and language) by reconstructing multiple neural network layer representations.

### 2.2 KNOWLEDGE DISTILLATION

Knowledge distillation (KD) is widely employed in model knowledge compression (Hinton et al., 2015), which improves the performance of the smaller student model by distilling the knowledge learned from a well-trained large teacher network. Further study on *e.g.*relational KD (Park et al.,

| computation pipeline | initialized teacher | classification | | | | object detection | | | | | semantic segmentation | | | | |
|---|---|---|---|---|---|---|---|---|---|---|---|---|---|---|---|
| | | 0th | 1st | 2nd | 3rd | 0th | 1st | 2nd | 3rd | 4th | 0th | 1st | 2nd | 3rd | 4th |
| Supervised | DeiT | 81.8 | 83.6 | 84.3 | 84.3 | 49.1 | 50.5 | 52.5 | 52.4 | - | 46.4 | 49.2 | 50.4 | 49.9 | - |
| Contrastive | DINO | 83.2 | 84.2 | 84.5 | 84.4 | 50.1 | 52.5 | 52.9 | 52.7 | - | 46.8 | 49.7 | 50.4 | 49.4 | - |
| Autoregressive | DALL-E | 81.1 | 83.5 | 84.4 | 84.3 | 31.9 | 51.0 | 52.7 | 52.5 | - | 31.9 | 47.4 | 49.6 | 49.3 | - |
| Autoencoding | MAE | 83.6 | 84.3 | 84.4 | 84.3 | 50.6 | 52.9 | 52.7 | 52.5 | - | 48.1 | 49.6 | 50.4 | 49.8 | - |
| - | random | 77.3 | 83.4 | 84.5 | 84.3 | 29.2 | 49.6 | 52.4 | 52.7 | 52.4 | 25.7 | 47.0 | 49.1 | 49.5 | 49.5 |
| performance variance | | 2.24 | 0.37 | 0.07 | 0.04 | 9.54 | 1.23 | 0.17 | 0.12 | - | 9.19 | 1.15 | 0.54 | 0.23 | - |

Table 1: The top-1 classification accuracy on ImageNet-1K, object detection AP-box on COCO with Cascade Mask R-CNN, and semantic segmentation mIoU on ADE20K with UperNet of dBOT using different models as the initialized teacher network. Note that all models are pre-trained on ImageNet-1K, including DALL-E, for a fair comparison. We perform distillation in each stage for 800 epochs. In the 1st stage, we distill from initialized teacher to obtain a student. In the subsequent (*i.e.*, 2nd, 3rd, etc.) stages, the obtained students are leveraged as bootstrapped teacher to distill a new student.

2019), contrastive KD (Tian et al., 2019), and latent feature KD (Romero et al., 2015) is conducted to improve the performance of vanilla KD. Beyond its prominence in the field of supervised learning, KD recently cuts a figure in self-supervised learning. Concurrent work manages to adopt conventional feature distillation (Wei et al., 2022c) to match contrastive models with MIM-trained ones. Nevertheless, it shows negligible gains on MIM-trained models such as MAE. BEiT (Bao et al., 2022), MaskFeat (Wei et al., 2022a) and MVP (Wei et al., 2022b) could be seen as distilling knowledge from dVAE (Ramesh et al., 2021), HOG features (Dalal & Triggs, 2005) and language-induced model CLIP (Radford et al., 2021) within the discourse of MKD, respectively. Until now, there exists no work conferring a system-level study on the importance of how to choose adequate target representation or teacher networks to guide the learning of MKD.

## 3   DOES $h_\phi(\cdot)$ MATTER IN MKD?

Given the general form of masked knowledge distillation as shown in Eq. (1), in this section, we aim to investigate *whether the careful design of the target,* i.e., *teacher network $h_\phi(\cdot)$, matters.* Specifically, we want to answer three questions as follows:

- Whether models distilled from different $h_\phi(\cdot)$ differ in terms of their transfer performances?
- Whether distilled models differ in terms of their weights and outputs?
- If $h_\phi(\cdot)$ does not matter, what matters more to close the gap between students distilled from different $h_\phi(\cdot)$?

To answer these questions, we employ the standard masked autoencoder framework (He et al., 2022) to give a system-level study, introduced next.

**Common setup.**   The architectural settings strictly follow  (He et al., 2022). For the teacher network, we use the vanilla ViT (Dosovitskiy et al., 2021) with intact input. For the student network with masked input, we use the asymmetric encoder-decoder structure. The student's output is further projected to a dimension the same as that of teacher's embedding. During pre-training, we use Smooth L1 loss (Girshick, 2015) for the optimization of the student network, and the teacher network is kept fixed. Detailed settings are delayed to Appendix A.1. We pre-train models on ImageNet-1K (Deng et al., 2009) and conduct evaluation under classification on ImageNet, object detection on COCO (Lin et al., 2014), and semantic segmentation on ADE20K (Zhou et al., 2017).

### 3.1   PRELIMINARY STUDY

We first investigate the effect of using networks initialized differently as teachers for masked knowledge distillation. Four canonical methods as ***pre-trained teachers*** are substantiated, each from a category distinguished based on their computation pipelines, *i.e.*, DeiT (Touvron et al., 2021) for

supervised learning, DINO (Caron et al., 2021) for contrastive learning, DALL-E (Ramesh et al., 2021) for autoregressive generation, and MAE (He et al., 2022) for autoencoding. The results of initialized teacher at the $0^{\text{th}}$ stage and of its distilled student at the $1^{\text{st}}$ stage are shown in Table 1.

**Different $h_\phi(\cdot)$ lead to similarly performed students.** After the first stage of masked knowledge distillation, the student consistently outperforms teacher as shown in Table 1, yielding 1.8%, 1.0%, 2.4%, and 0.7% performance gains for four different $h_\phi(\cdot)$ respectively, demonstrating the effectiveness of masked knowledge distillation for visual representation learning. Although the performance order of different $h_\phi(\cdot)$ is reserved after the first stage of distillation, the students distilled from different $h_\phi(\cdot)$ have closer downstream performances compared to the original $h_\phi(\cdot)$. The performance variance drops from 2.24 to 0.37 after the first stage of distillation. The conclusion holds true for experiments on object detection and semantic segmentation.

## 3.2 DISTILLATION WITH MULTIPLE STAGES

Given the observations that better teacher generally induces better outperforming student, we are motivated to use the trained student as teacher to train new student repeatedly and study whether similar trend endures. If so, we would like to seek at what stage the performances saturate for different downstream tasks, as well as the discrepancy among the results incurred by different initialized teachers.

**$h_\phi(\cdot)$ does not matter with multi-stage distillation.** The performance gain is valid but decreases with multi-stage and eventually vanishes. Take MAE being the initialized teacher as an example, students outperform teachers by +0.7%, +0.1%, -0.1% for classification, +2.3, -0.2, -0.2 points for object detection, and +1.5, +0.8, -0.6 points, for semantic segmentation, from the $0^{\text{th}}$ to the $3^{\text{rd}}$ stage. Other teachers and downstream tasks share the same conclusion. Moreover, the performance gaps of students learned from different teachers decrease, especially after multi-stage, as shown by the performance variance at different stages in the last row of Table 1. Take classification tasks for instance, the variance decreases along with the training stage, *i.e.*, 2.24, 0.37, 0.07, 0.04, which reveals that the choice of $h_\phi(\cdot)$ exerts little influence on the downstream performance. See Table 1 for results of more downstream tasks. To demonstrate models' differences in terms of weights and outputs, we conduct a property analysis in Sec. 6. Similar properties are found, which verify our conclusion.

**A random $h_\phi(\cdot)$ works surprisingly well.** Since the choice of $h_\phi(\cdot)$ does not matter, an intuitive experiment is to see what will happen when we employ a ***random teacher***, in which the parameters are randomly initialized at the $0^{th}$ stage. To our surprise, using a random teacher achieves performances comparably with other pre-trained teachers. Compared to a randomly initialized model, distilled students with multiple stages achieve 6.1%, 20.4, and 21.3 performance gain on classification, object detection and semantic segmentation respectively. Empirically, object detection and semantic segmentation require one more stage to saturate compared to classification. The saturated results are on par with those induced by pre-trained teachers, which enables us to train a state-of-the-art model more efficiently, without the need of an extra pre-training stage for the initialized teacher (*e.g.*, contrastive learning as DINO).

## 4 MKD WITH BOOTSTRAPPED TEACHERS

The study in Sec. 3 motivates us to propose a multi-stage distillation pipeline for pre-training. The entire pre-training undergoes multiple stages split by breakpoints. For each stage, we fix teacher network to obtain a stable visual representation, guiding the learning of student network. The pre-trained student model is then used as a stronger teacher and distills its knowledge to a new subsequent student, providing richer visual representations. We re-initialize the student network at each breakpoint. The above process repeats itself - the teachers keep bootstrapped from the students, until a performance saturation on downstream tasks is observed. Hence, our strategy is to perform distillation with *bootstrapped teacher*s. We illustrate our framework in Fig. 1c and the conceptual relations with the other two paradigms in Fig. 1. By noting $m$ as the momentum which indicates how fast the teacher's parameters $\boldsymbol{\theta}_t$ is updated from student's parameters $\boldsymbol{\theta}_s$, *i.e.*, $\boldsymbol{\theta}_t = m \cdot \boldsymbol{\theta}_t + (1 - m) \cdot \boldsymbol{\theta}_s$, we present the following discussions.

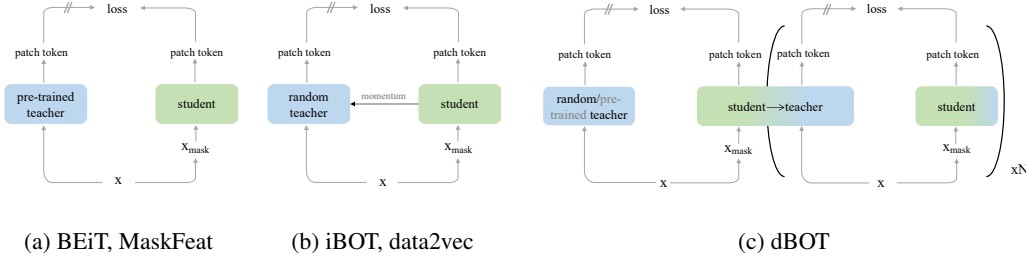

(a) BEiT, MaskFeat      (b) iBOT, data2vec      (c) dBOT

Figure 1: **Conceptual comparison of three masked image modeling paradigms.** The difference between the three paradigms is how the parameters of the teacher network are updated. (a): The parameters of the teacher network are frozen during the whole training process, constructing an offline teacher. (b): Exponential moving average is applied to correlate the parameters of the student and teacher networks, constructing an online teacher. (c): dBOT uses a multi-stage distillation pipeline, *i.e.*, the parameters of the teacher network are frozen except at breakpoints, where we assign student parameters to the teacher and re-initialize the student network.

**Relations with previous methods.** One group of works leverages ***pre-trained teacher*** as in Fig. 1a, *i.e.*, BEiT (Bao et al., 2022). The teacher requires an extra stage of pre-training and is kept fixed with $m = 1$. Ideally, pre-trained teachers bear additional knowledge which is prone to be more semantic meaningful, prompting student's learning. Nonetheless, the pre-training of these teachers entails a completely different computation pipeline (Wei et al., 2022a) and often additional data (Wei et al., 2022b), complicating its practical use. Another group as in Fig. 1b works with ***random teacher*** in dispense with pre-trained ones. Starting from randomness, the teachers in iBOT (Zhou et al., 2021) and data2vec (Baevski et al., 2022), however, are bootstrapped from the student typically with $m \in (0, 1)$, *e.g.*, 0.9998 as in (Baevski et al., 2022). Although bootstrap induces improving quality of the teacher's representation, the pipeline is plagued by its optimization instability and sensitivity towards hyper-parameters. We note that MAE uses identity mapping of pixels as the target, which is observed to function similarly as a fixed random teacher with $m = 1$, as shown in Appendix B.1. Despite its simplicity, such practice eludes synergy between the teacher and the student. Comparatively, dBOT is with $m = 0$ for every breakpoint and $m = 1$ otherwise.

## 5 EXPERIMENTS

### 5.1 PRE-TRAINING

**Architecture.** We use different capacity Vision Transformers (Dosovitskiy et al., 2021), *i.e.*, ViT-B/16, ViT-L/16, and ViT-H/14 for dBOT. The input image of size 224×224 is first divided by a linear projection head into non-overlapping patch tokens total of 196 for ViT-B and ViT-L, and 256 for ViT-H. We exactly follow the common setup demonstrated in Sec. 3, *e.g.*, a student with asymmetric encoder-decoder architecture, a teacher with intact input, etc.

**Optimization.** The learning rate is first linearly increased to the initial learning rate for the first 40 epochs and then cosine annealed to 0. The initial learning rate is set as 1.5e-4 × batch_size / 256, with batch size being 4096 for all models. We use the AdamW optimizer (Loshchilov & Hutter, 2019) and Smooth L1 loss (Girshick, 2015) to optimize the parameters of student network. Stochastic drop rate are applied, 0.2 for ViT-B, 0.2 for ViT-L, and 0.3 for ViT-H. We use only center-crop and flipping for data augmentation. As shown in Table 1, the performance of different downstream tasks saturates at different stages. By default, we pre-train all models for classification with 2 stages, for object detection and semantic segmentation with 3 stages.

### 5.2 IMAGENET RESULTS

We primarily focus on the end-to-end fine-tuning performance and report the top-1 validation accuracy on ImageNet-1K (Deng et al., 2009) dataset.

Table 2: **Comparison fine-tuning result of the previous methods on ImageNet-1K.** We evaluate by the end-to-end fine-tuning protocol. All results are based on an image size of 224, except for ViT-H with an extra result with 448 image size. We perform distillation in each stage for 800 epochs and with 2 stages (our default) in total.

| method | ViT-B | ViT-L | ViT-H | ViT-H$_{448}$ |
|---|---|---|---|---|
| supervised | 82.3 | 82.6 | 83.1 | - |
| MoCo v3 | 83.2 | 84.1 | - | - |
| DINO | 83.6 | - | - | - |
| *methods based on masked image modeling:* | | | | |
| BEiT | 83.2 | 85.2 | - | - |
| iBOT | 84.0 | 85.2 | - | - |
| MAE | 83.6 | 85.9 | 86.9 | 87.8 |
| data2vec | 84.2 | 86.2 | - | - |
| dBOT | **84.5** | **86.6** | **87.4** | **88.0** |

Table 3: **Semi-supervised learning on ImageNet-1K** with different self-supervised models. 1% and 10% represent the label fraction. ViT-B is selected as the arch. All results are based on our implementation with the official pre-trained model.

| method | 1% | 10% |
|---|---|---|
| supervised | - | 68.9 |
| data2vec | 48.7 | 71.2 |
| MAE | 53.1 | 73.1 |
| dBOT | **54.8** | **74.5** |

Table 4: **Object detection and instance segmentation on COCO and Semantic segmentation on ADE20K.** All results are based on our implementation with the official pre-trained model. We perform distillation in each stage for 800 epochs and with 3 stages (default).

| method | AP$^{box}$ | | AP$^{mask}$ | | method | mIoU | | mAcc | |
|---|---|---|---|---|---|---|---|---|---|
| | ViT-B | ViT-L | ViT-B | ViT-L | | ViT-B | ViT-L | ViT-B | ViT-L |
| supervised | 49.8 | 51.2 | 43.2 | 44.5 | supervised | 47.4 | 49.9 | - | - |
| DINO | 50.1 | - | 43.4 | - | iBOT | 48.4 | 52.3 | 59.3 | 63.3 |
| MAE | 50.6 | 54.0 | 43.9 | 46.2 | data2vec | 48.2 | - | 59.5 | - |
| iBOT | 51.3 | - | 44.3 | - | MAE | 48.1 | 53.6 | 58.9 | 65.5 |
| dBOT | **52.7** | **56.0** | **45.7** | **48.2** | dBOT | **49.5** | **54.5** | **60.7** | **66.0** |

**Evaluation setup.** We sweep the base learning rate within a range with a batch size being 1024. We warm up the learning rate during the first 5 epochs to the initial learning rate and use a cosine schedule for the rest of the epochs. We average all the patch tokens output from the last transformer block and pass them into a linear projection head for classification. We fine-tune ViT-B for 100 epochs and ViT-L and ViT-H for 50 epochs in total.

**Comparison with previous results.** We report the fine-tuning results on ImageNet-1K, mainly focusing on the comparison of the self-supervised and supervised methods. Supervised denotes the results reported in the MAE. As shown in Table 2, dBOT achieves remarkable results with different model capacities, demonstrating its scalability. We achieved top-1 evaluation accuracy of 84.5%, 86.6%, and 87.4% with ViT-B, ViT-L, and ViT-H, yielding gains of 0.9%, 0.7%, and 0.5% compared to MAE. When fine-tuned with an image size of 448, dBOT further achieves an accuracy of 88.0%, surpassing the results obtained by MAE.

**Semi-supervised learning.** To investigate the label efficiency of dBOT, we also show the semi-supervised results on ImageNet-1K under different labeled data availability in Table 3. We focus on the comparison with self-supervised learning methods. The label-fraction sampling strategy follows (Chen et al., 2020). dBOT outperforms MAE by 1.7 and 1.4 points using 1% and 10% of the labels, respectively, showing a higher label efficiency.

## 5.3 DOWNSTREAM TASKS

To further demonstrate the effectiveness, we consider dense prediction tasks: object detection, semantic segmentation, and instance segmentation.

**Objection detection and instance segmentation.** We consider Cascade Mask R-CNN (Cai & Vasconcelos, 2019) as the task head for object detection and instance segmentation with ViT-B and ViT-L on COCO (Lin et al., 2014). We report AP$^{box}$ and AP$^{mask}$ for object detection and instance segmentation respectively. The results are demonstrated in Table 4. dBOT outperforms the

Table 5: **Ablation study** with **ViT-B/16** on ImageNet-1K validation set. We report with the end-to-end fine-tuning top-1 accuracy (%). Ablation study is conducted with randomly initialized teachers. We note that models distilled from the pre-trained teachers generally share similar trends. Default settings are marked in ​ gray ​. *vanilla* denotes $m$ being 0 at the breakpoint and 1 otherwise. cosine(a,b) denotes $m$ is cosine annealed from value a to b.

(a) **Stage split number**. 2-stage distillation works the best.

| pre-training epochs | acc |
|---|---|
| 1600 | 83.6 |
| 800-800 | **84.5** |
| 533-533-533 | 84.4 |

(b) **Epoch for each stage**. 2-stage distillation with 800 epochs for each stage works the best.

| pre-training epochs | acc |
|---|---|
| 400-800 | 84.3 |
| 800-400 | 84.3 |
| 800-800 | **84.5** |
| 800-1200 | 84.3 |

(c) **Momentum update**. The *vanilla* strategy explicitly splitting stages works the best.

| momentum | acc |
|---|---|
| *vanilla* | **84.5** |
| 0.9998 | 83.6 |
| 0.9999 | 83.9 |
| cosine(0.996,1) | 82.1 |

(d) **Target normalization**. Using patch representations w/o [LN] as targets works best.

| target norm | acc |
|---|---|
| w/ [LN] | 84.3 |
| w/o [LN] | **84.5** |

(e) **Student initialization**. Re-initializing the student's weight at breakpoints works best.

| student init | acc |
|---|---|
| w/o re-initialize | 84.2 |
| w/ re-initialize | **84.5** |

(f) **Mask ratio**. A mask ratio of 75% works best.

| mask ratio | acc |
|---|---|
| 0.7 | 84.3 |
| 0.75 | **84.5** |
| 0.8 | 84.2 |

previous self-supervised and supervised methods by a large margin, setting a new state-of-the-art result with both ViT-B and ViT-L. With ViT-B, dBOT achieves a $AP^{box}$ of 52.7 and a $AP^{mask}$ of 45.7, outperforming the supervised baseline pre-training by 2.9 and 2.5 points, respectively. With ViT-L, such improvement is more prominent with 4.8 and 3.6 points respectively, showing the high scalability of dBOT for model capacity in downstream dense prediction tasks.

**Semantic segmentation.** We adapt UperNet (Xiao et al., 2018) as the task head for semantic segmentation with ViT-B and ViT-L on ADE20K (Zhou et al., 2017). We report the mIoU and mAcc for semantic segmentation, and the results are demonstrated in Table 4. We achieve the best performances on semantic segmentation compared to previous self-supervised methods by a nontrivial margin. dBOT improves mIoU from 47.4 to 49.5 with ViT-B, and 49.9 to 54.5 with ViT-L, yielding gains of 2.1 and 4.6 points respectively, compared to the supervised baseline. The improvement in semantic segmentation is as significant as in object detection.

## 5.4 ABLATION STUDY

**Stage split number.** We study the influence of stage number by splitting total training epochs of 1600 into varying distillation stages, from 0 to 2. Results are shown in Table 5a. 2-stage distillation works the best (for classification task), achieving 84.5% accuracy. Splitting epochs to 3-stage brings 0.1% performance drop, while all splitting strategies obtain a top-1 accuracy higher than 83.6%, indicating its generalizability.

**Epoch for each stage.** Table 5b studies proper epochs needed for each stage in a 2-stage distillation pipeline. With the 2[nd] stage distilling for 800 epochs, longer epochs for the 1[st] stage induces 0.2% improvement (84.3% *vs.* 84.5%). With the 1[st] stage distilling for 800 epochs, 800 epochs are enough for the 2[nd] stage since 1200 epochs incur no gain. Evenly splitting the epochs in 2-stage masked knowledge distillation achieves the best performance.

**Momentum update.** We use in dBOT a multi-stage distillation pipeline, which is to distill from a momentum encoder with $m$ being 0 for every breakpoint and 1 otherwise. We further investigate other momentum update strategies commonly used in self-supervised learning. Results are shown in Table 5c. The *vanilla* strategy works the best.

**Target normalization.** We study whether patch tokens obtained by the self-attention blocks to be used as target representation should be passed through the Layer Normalization (Ba et al., 2016)

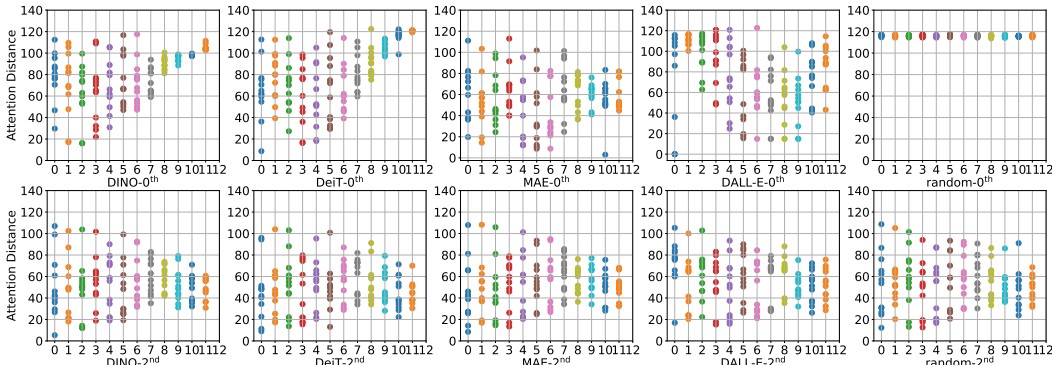

Figure 2: Average attention distance of different heads w.r.t layer number of ViT-B with different teachers and their corresponding student distilled for 2 stages. The first row showcases the teachers while the second showcases the $2^{th}$ stage distilled student. Models using different teachers achieve the same result. The distilled students obtain more local attention compared to the teachers.

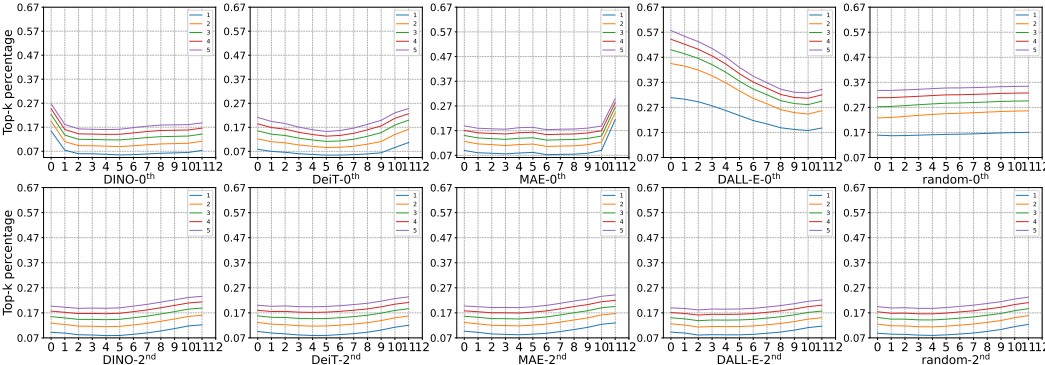

Figure 3: Singular value decomposition of different layers of ViT-B with different teachers and their corresponding student distilled for 2 stages. The first row showcases the teachers while the second showcases the $2^{th}$ stage distilled student. Models using different teachers achieve the same result.

layer [LN]. The accuracy of models after 2-stage distillation is shown in Table 5d. Without passing through [LN], the patch tokens directly obtained from the transformer block make them less suitable as target representations to guide students' learning.

**Student initialization.** We study whether student's weight should remain when entering the next stage of distillation. Specifically, we either keep the student's weight unchanged or re-initialize the student at each breakpoint. As shown in Table 5e, re-initializing the student's weight works the best.

**Mask ratio.** Table 5f shows the influence of the mask ratio on end-to-end fine-tuning. The optimal mask ratio for dBOT is 75%, the same as that in MAE.

## 6 PROPERTY ANALYSIS

We investigate the properties of models distilled from different teachers under certain criteria, analyzing models' weights and outputs. Further, training efficiency is briefly discussed with previous methods.

**Averaged attention distance.** We compute averaged attention distance (Dosovitskiy et al., 2021), averaged over ImageNet-1K val set, for each attention head of different blocks to understand how local and global information flows into Transformers. Average attention distance for dBOT using DeiT, DINO, MAE, DALL-E, and random as teachers are illustrated in Fig. 2. The higher the

Table 6: **Training time (s) per epoch for different methods with ViT-B/16, ViT-L/16, and ViT-H/14.** *asym.* denotes whether to use an asymmetric encoder-decoder structure. All entries are tested on the same setting, *i.e.*, with 32 NVIDIA A100-80G GPUs.

| method | data2vec | BEiT | MAE | dBOT |
|--------|----------|------|-----|------|
| *asym.* | ✗ | ✗ | ✓ | ✓ |
| ViT-B | 169 | 166 | 79 | 109 |
| ViT-L | 431 | 356 | 125 | 200 |
| ViT-H | 960 | 751 | 240 | 416 |

Table 7: **Results of classification (cls.) on IN1K, object detection (det.) on COCO, and semantic segmentation (seg.) on ADE20K.** For same-size teachers (colored gray), students are pre-trained with default settings. For bigger teachers, students are pre-trained for 1-stage from 2-stage distilled teachers.

| teacher | student | cls. | det. | seg. |
|---------|---------|------|------|------|
| ViT-B | | 84.5 | 52.7 | 49.5 |
| ViT-L | ViT-B | 84.6 (+0.1) | 53.1 (+0.4) | 50.1 (+0.6) |
| ViT-H | | 84.6 (+0.1) | 53.5 (+0.8) | 50.8 (+1.3) |
| ViT-L | | 86.6 | 56.0 | 54.5 |
| ViT-H | ViT-L | 86.8 (+0.2) | 56.1 (+0.1) | 55.2 (+0.7) |

attention distance, models' attention over an image is more global. Although the average attention distance of disparate initialized teachers varies greatly, their distilled students after multi-stage distillation exhibit similar behaviors, *e.g.*, models' attention toward local or global contents. Additionally, dBOT achieves more local attention than previous works.

**Singular value decomposition.** We computed the percentage of top-$k$ singular values (Wall et al., 2003) of the embedding w.r.t each layer. The results are averaged over the ImageNet-1K val set. We showcase the results with $k$ varying from 1 to 5. Singular value decomposition for dBOT using DeiT, DINO, MAE, DALL-E, and random as teachers are shown in Fig. 3. The higher the percentage, the models' output over an image is less correlated, indicating larger redundancy of its spatial representations thus less suitability for compression. Intuitively, random models at the $0^{th}$ stage has the largest percentage given that pixel are merely randomly projected. The student networks distilled from different initialized teachers exhibit similar behaviors.

**Training efficiency.** We compute the training time per epoch for different methods in Table 6. With an asymmetric encoder-decoder architecture (*asym.*) as the default setup, dBOT performs slower than MAE, but much faster than data2vec and BEiT. Such advantage turns more significant with models of larger size.

## 7 DISTILL FROM BIGGER TEACHERS

Inspired by canonical practices in knowledge distillation (Hinton et al., 2015), we use larger teachers to distill smaller students, showcasing the potential of MKD in general. Specifically, we attempt to use ViT-L/H as teacher networks to distill ViT-B, and ViT-H as the teacher network to distill ViT-L. All larger teachers are first distilled for 2 stages with the default setup. We resize the image to 196×196 for ViT-H/14 to keep the length of its output the same as that of ViT-B/L. While we do not find substantial gains on classification results, the results by distilling from ViT-H are significantly better for dense prediction tasks compared to the default setup, *i.e.*, +0.8 points of $AP^{box}$ and +1.3 points of mIoU with ViT-B as the student. The performance gain in distilling ViT-L from ViT-H is diminished but still valid, *i.e.*, +0.1 $AP^{box}$ and +0.7 mIoU. We also consider MKD with data-richer teachers, *e.g.*CLIP, as exploratory experiments and set new state-of-the-art results for self-supervised learning. Refer to Appendix C for details.

## 8 CONCLUSION

As a special case of MIM, we formulate MKD upon which an empirical investigation is conducted about the influence of different target representations on self-supervised masked autoencoders. The study concludes that it is *not necessary* to carefully choose the target representation to learn good visual representations if distillation is performed in multiple stages (*i.e.*, with bootstrapped teachers). Instead of initializing teachers with pre-trained models, we resort to random ones for simple practice. *Without an extra stage of pre-training*, dBOT achieves favorable performance on image classification, object detection, and semantic segmentation. We hope our study and method will provide timely insights for self-supervised learning.

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

Table A1: **Pre-training setup.** recipe🏃 is the pre-training recipe for dBOT🏃. cos. denotes cosine distance. c., d., and s. denotes downstream tasks of classification, object detection, and semantic segmentation respectively. drop path is for the students.

| config | default | recipe🏃 |
|---|---|---|
| optimizer | AdamW | |
| optim. momentum $\beta_1$ | 0.9 | |
| optim. momentum $\beta_2$ | 0.95 | 0.98 |
| loss | Smooth L1 | negative cos. |
| peak learning rate | 2.4e-3 | 3e-3 |
| learning rate schedule | cosine decay | |
| batch size | 4096 | |
| weight decay | 0.05 | |
| stages | 2 (c.), 3 (d./s.) | 1 |
| epochs per stage | 800 | 1600 |
| warmup epochs | 40 | 10 |
| augmentation | RandomResizedCrop | |
| aug. input scale | (0.2, 1) | (0.4, 1) |
| asym. enc-dec | ✓ | ✗ |
| drop path | 0.2 (B/L), 0.3 (H) | 0.1 (B/L/H) |
| target w/ [LN] | ✗ | ✓ |
| mask ratio | 0.75 | 0.4 |

Table A2: **End-to-end fine-tuning setup.** recipe🏃 is the pre-training recipe for dBOT🏃.

| config | default | recipe🏃 |
|---|---|---|
| optimizer | AdamW | |
| peak learning rate | {0.8,1.2,1.6,2}e-3 | {1,2,3,4}e-4 |
| weight decay | 0.05 | |
| optim. momentum | $\beta_1, \beta_2 = 0.9, 0.999$ | |
| layer-wise decay | 0.75 | |
| batch size | 1024 | |
| learning schedule | cosine decay | |
| warmup epochs | 5 | |
| epochs | 100 (B), 50 (L/H) | |
| augmentation | RandAug (9, 0.5) | |
| label smoothing | 0.1 | |
| mixup | 0.8 | |
| cutmix | 1.0 | |
| drop path | 0.2 (B/L), 0.3 (H) | 0.1 (B), 0.2 (L), 0.3 (H) |

# A    IMPLEMENTATION DETAILS

## A.1    PRE-TRAINING

**Default setup.**    We show our *default* pre-training setup in the second colum of Table A1. We use Xavier Uniform (Glorot & Bengio, 2010) to initialize the Vision Transformer (Dosovitskiy et al., 2021). Note that we use asymmetry stochastic drop path rate for students and teachers.

**Setup for distillation from bigger teachers.**    We follow the *default* setup, except that we use a different setup for stages. We first train larger-size teachers for 2 stages (in all downstream tasks) and use those to distill new students for 1 stage (in all downstream tasks).

Table A3: **Object detection setup.**

| config | value |
|---|---|
| optimizer | AdamW |
| optim. momentum | $\beta_1, \beta_2 = 0.9, 0.999$ |
| peak learning rate | 1e-4 |
| batch size | 16 |
| layer-wise decay | 0.75 |
| weight decay | 0.05 |
| learning schedule | step |
| epochs | 12 |
| step epochs | 8, 11 |
| drop path | 0.2 |

Table A4: **Semantic segmentation setup.**

| config | value |
|---|---|
| optimizer | AdamW |
| optim. momentum | $\beta_1, \beta_2 = 0.9, 0.999$ |
| peak learning rate | {0.3,0.5,0.8,1,3}e-4 |
| batch size | 16 |
| layer-wise decay | {0.65,0.75,0.85.0.95} |
| weight decay | 0.05 |
| learning schedule | cosine |
| steps | 16000 |
| warmup steps | 1500 |
| drop path | 0.1(B), 0.2(L) |

## A.2 CLASSIFICATION

The *default* end-to-end fine-tuning recipe is shown in the second column of Table A2, following the common recipes (He et al., 2022; Bao et al., 2022) of ViT tuning for self-supervised models. The same recipe is applied when distilling from bigger teachers.

## A.3 OBJECT DETECTION AND INSTANCE SEGMENTATION

We adopt the vanilla ViT with Cascade Mask R-CNN (Cai & Vasconcelos, 2019) as the task head on COCO (Lin et al., 2014) dataset for object detection and instance segmentation, following the common setup (Zhou et al., 2021). The default recipe is shown in Table A3. To cope with versatile image sizes, we add relative position embedding instead of interpolating the absolute position embedding obtained during pre-training. For a fair comparison, we applied the same setup and sweep the learning rate and stochastic drop path rate for different methods.

## A.4 SEMANTIC SEGMENTATION

We use vanilla ViT and UperNet (Xiao et al., 2018) as the task head on ADE20K (Zhou et al., 2017) dataset for semantic segmentation, following the common setup (Bao et al., 2022). The default recipe is shown in Table A4. To cope with versatile image sizes, we add relative position embedding instead of interpolating the absolute position embedding obtained during pre-training. For a fair comparison, we applied the same setup and sweep the learning rate and layer-wise decay for different methods.

## B ADDITIONAL EXPERIMENTS

### B.1 PIXELS *vs.* RANDOM MAPPING OF PIXELS

MAE performs masked image modeling using the image pixel as the reconstruction target. We directly alter the target to patch tokens obtained from the image fed into a randomly initialized network. We select two patch tokens as the reconstruction target, one is the token obtained using the last transformer block, and the other is the token obtained using linear projection, *i.e.*, without any transformer block. After 400 epoch pre-training of ViT-B, the top-1 accuracy of the model on ImageNet-1K obtained by the three different targets is shown below.

| epoch | pixel | $0^{\text{th}}$ block | $12^{\text{th}}$ block |
|---|---|---|---|
| 400 | 83.3 | 83.2 | 83.2 |
| 1600 | 83.6 | 83.6 | 83.6 |

It can be derived that using the patch token obtained by a randomly initialized network as the target can achieve comparable results with a pixel as a target. A similar result proves that patch tokens obtained by a randomly initialized can also serve as a good reconstruction target.

## B.2 Linear Probing

We evaluate the linear probing performance of dBOT and MAE using ViT-B following the same setup as MAE, the results of which is shown below.

| MAE | dBOT |
|-----|------|
| 67.8% | 67.9% |

dBOT achieves comparable linear probing performances with MAE.

## C Distill from Data-Richer Teachers

We explore to use models pre-trained with richer data (*i.e.*, CLIP (Radford et al., 2021) with 400M Image-Text pairs) as the initialized teacher to seek a potential upper-bound of MKD.

### C.1 Pre-Training

Compared to the *default* setup, there exist two major disparities of the pre-training recipes for models distilled from data-richer teachers, discussed next. The following practice is summarized as *recipe* detailed in Table A1.

**Vanilla Architecture.** We find that not using the asymmetric encoder-decoder architecture (He et al., 2022) is optimal, as shown in Table C5. While an asymmetric architecture generates momentum for bootstrapping models similar to (Grill et al., 2020), which lies crucial for distillation with random teachers, it hurts the performance when distilling with stronger pre-trained teachers.

Hypothetically, the significance of the decoder in asymmetrical encoder-decoder architecture lies in the need for separate layers to decode low-level details when the targets contain little semantics (*e.g.*, pixels and random mappings of pixels). Such a need is eased when the target contains high-level semantics (*e.g.*, DINO and CLIP). The existence of the decoder, in this case, may even restrain the encoder to grasp full knowledge from the teacher, inducing degraded performances.

**1-Stage MKD**. We use different models as teachers to distill students for one stage with longer epochs, *i.e.*, 1600. Results are shown in Table C6. Empirically, the performance gains for multi-stage MKD over 1-stage MKD decrease as teachers' fine-tuning performance increases. Stronger teachers, such as DINO and MAE, induce similarly performed students with 1-stage MKD ($1\times1600$) compared to 2-stage MKD ($2\times800$).

Specifically, when using CLIP as the pre-trained teacher, the performance for 2-stage MKD is, to our surprise, 0.9% lower than that of 1-stage MKD. Understandably, although the fine-tuning result of the student after 1-stage distillation is better than that of CLIP, the student is essentially trained on IN1K and may not contain faithfully data information stored in the CLIP model. Therefore, strong teachers work well with 1-stage MKD, especially for models pre-trained on extra richer data.

### C.2 Downstream Tasks

**Implementation Details.** For fine-tuning, we also use a slightly different recipe from *default* one with smaller learning rates and drop path, dubbed as *recipe* detailed in Table A2. For object

Table C5: Image classification on IN1K with DINO and CLIP as initialized teachers, as well as random ones. Students with DINO and CLIP as teachers are distilled for 1 stage.

| initialized teacher | pre-training data | asym. enc-dec | acc |
|---------------------|-------------------|---------------|------|
| random | IN1K | ✓ | 84.5 |
| random | IN1K | ✗ | 83.8 |
| DINO | IN1K | ✓ | 84.4 |
| DINO | IN1K | ✗ | 84.8 |
| CLIP | IN1K + 400M ITp. | ✓ | 84.9 |
| CLIP | IN1K + 400M ITp. | ✗ | 85.7 |

Table C6: ImageNet-1K classification results of 1 stage masked knowledge distillation with different teachers. Total epochs are shown in the format of (stages×epochs_per_stage). △ denotes performance gaps between entries of 2×800 and 1×1600.

| pretraining epochs | random | DALL-E | DeiT | DINO | MAE | CLIP |
|---|---|---|---|---|---|---|
| 0 | 77.3 | 81.1 | 81.8 | 83.2 | 83.6 | 84.8 |
| 1×1600 | 83.6 | 83.6 | 83.6 | 84.4 | 84.4 | 84.9 |
| 2×800 | 84.5 | 84.4 | 84.3 | 84.5 | 84.4 | 84.0 |
| △ | +0.9 | +0.8 | +0.7 | +0.1 | +0.0 | -0.9 |

Table C7: Results of classification (cls.) on IN1K, object detection (det.) on COCO, and semantic segmentation (seg.) on ADE20K with CLIP (Radford et al., 2021) as the teacher. Students are distilled for 1 stage. The det. results with CLIP as teachers are with absolute positional embedding.

| initialized teacher | student | cls. | det. | seg. |
|---|---|---|---|---|
| random | ViT-B | 84.5 | 52.7 | 49.5 |
| CLIP-B | | 85.7 (+1.2) | 53.6 (+0.9) | 52.9 (+3.4) |
| random | ViT-L | 86.6 | 56.0 | 54.5 |
| CLIP-L | | 87.8 (+1.2) | 56.8 (+0.8) | 56.2 (+1.7) |
| random | ViT-H | 87.4 | - | - |
| CLIP-L | | 88.5 (+1.1) | - | - |

detection, instance segmentation, and semantic segmentation, we follow the default setup detailed in Appendices A.3 and A.4.

**Results.** Results for downstream tasks are shown in Table C7. ViT-B distilled from CLIP-B achieves an 85.7% top-1 accuracy and a 52.9 mIoU, surpassing all previous arts. With CLIP-L as the teacher, ViT-H with image resolution 448 achieves an **89.1%** top-1 accuracy, setting a new state-of-the-art image recognition result.

## C.3 CONFLICT WITH MAIN CONCLUSION

It can be observed that MKD with CLIP (Radford et al., 2021) as the teacher performs much better than that with the random teacher and multi-stage distillation, which seems contradictory to our main conclusion that *teacher networks do not matter with multi-stage masked knowledge distillation.* Notably, CLIP is trained with 400M image text pairs (300× larger than ImageNet-1K), which is a drastically different setup from multi-stage distillation on ImageNet-1K only. Exploring CLIP as a target representation gains popularity (Wei et al., 2022b) recently but is beyond the main scope of this paper. We present these results to corroborate the validity and to explore the upper bound of MKD in general. We note that the exact solution to resolve the conflict is to perform multi-stage distillation using the CLIP's in-house 400M data to which we have no access. It is hypothesized that two results should be matched in light of experiments on ImageNet-1K, which is left to future work.

