# OpenReview forum: "Exploring Target Representations for Masked Autoencoders"
_ICLR.cc/2024/Conference — ICLR 2024 poster_

### Official Review · Reviewer_wVuK · 2023-10-31

**Soundness:** 2 fair
**Presentation:** 3 good
**Contribution:** 2 fair
**Rating:** 6
**Confidence:** 3

**Summary:**

This paper explores a quite interesting perspective of masking image modeling by first summarizing current methods with a more high-level architecture called masking knowledge distillation and then empirically demonstrate the learnt backbones of different target representation do not differ from each other with respect to both transfer results and weight distribution. Based on that, the authors propose a simple yet effective framework called dBOT to learn strong self-supervised visual representation.

**Strengths:**

- This paper has a clear formulation and writing architecture to present their motivation.
- The experimental results are solid and completed to support their claim.
- The observed phenomenon is quite interesting.

**Weaknesses:**

- This paper constructs tightly with the proposed masked knowledge distillation framework in Equ. 1, which consists of several basic components, including 1) the transferred backbone, 2) target representation, 3) asymmetric masking and 4) similarity measurement.
- About target representation:
  - The phenomenon observed in this paper can be more detailed phrased as, "With long enough pre-training, different target representation demonstrate similar behavior". Therefore, does that suggest that the observed conclusion only mask sense with a fixed dataset with long enough training?
  - In other words, is the bottleneck of scalable visual representation learning not about the methodology but more about data?
- About asymmetric masking:
  - Following my question above, another perspective to understand this phenomenon is due to the limited scalability [1] of asymmetric masking architecture proposed in MAE, which is also utilized as the main architecture in this work. While in Swinv2 and EVA, the MIM pre-training has been quite important for ViT training of giant sizes. Does the implementation of masking also affect the scalability of pre-training?
  - Moreover, does the invariance of target representation suggest masking operation is the key in MKD, which might also explain why the learnt representation has similar weight distribution?
- About backbone architecture:
  - Throughout the whole paper, the authors utilize the vanilla ViT of different sizes for both the student and teacher networks, while DeiT has shown that architecture discrepancy exists when distilling  between different architectures like CNN and ViT. Will that be the same for MKD, like does the phenomenon still hold with a ConvNext teacher?
- Overall, I think this is a quite interesting paper, and my questions are more open-ended to further enhance the insight of the observed phenomenon for future development.

[1] Zhai, Xiaohua, et al. "Scaling vision transformers." *Proceedings of the IEEE/CVF Conference on Computer Vision and Pattern Recognition*. 2022.

**Questions:**

- Writing:
  - Typo: "self-supervised learning" instead of "self-supervised learninf" in Keywords

---

> ### Author Response · Authors · 2023-11-13
>
> Thanks for your valuable comments and your appreciation for our technical contributions. We will fix the typos. Below, we discuss in detail the points you have raised.
>
> **Q1: Is the bottleneck of scalable visual representation learning not about the methodology but more about data?**
>
> In our paper, we conducted experiments with the same dataset (IN-1K) and the same setting (multi-stage distillation). We specifically focused on masked knowledge distillation, which utilizes masked image modeling as the pretext task for self-supervised training. We observed that the target representation is not crucial for mask knowledge distillation when distillation is performed in stages. However, we believe that new sophisticated pretext tasks can further enhance the performance. The bottleneck of scalable visual representation learning arises from both methodology and data.
>
> **Q2: Does the invariance of target representation suggest that the masking operation is the key in MKD? Does the implementation of masking also affect the scalability of pre-training?**
>
> The masking operation is indeed the key in masked knowledge distillation for self-supervised learning. Through the same pre-training task, the target representation is not important in multi-stage distillation, and dBOT can achieve a similar weight distribution. The implementation of masking also affects the scalability of pre-training. Like MAE, even changing the mask ratio can affect the model performance, as shown in our ablation study.
>
> **Q3: More backbone architecture.**
>
> In our experiment, we used the same architecture for both the student and teacher networks, as the teacher's weights are loaded from the pre-stage student's weights in multi-stage distillation. Therefore, the teacher and student networks should be the same. Previous works such as DINO and DALL-E were pre-trained on vanilla VIT, which facilitated our study on masked knowledge distillation with vanilla VIT as the backbone. As you mentioned ConvNext, a possible experiment would involve performing DINO and DALL-E on ConvNext first, and then applying masked knowledge distillation. However, whether DINO and DALL-E are suitable for ConvNext is a question that goes beyond the scope of our paper. We would like to leave this as an open-ended question.
>
> We appreciate your valuable feedback and will continue to explore and expand the scope of our work.

---

> > ### Comment · Reviewer_wVuK · 2023-11-22
> > **Response to Author Reply**
> >
> > Thanks for your respones. However, my understanding about this paper actually remains the same with my pre-rebuttal opinion. I do agree with Reviewer aTJx that the phenomenon of "target representation not important" is probably just another "fancy story" to demonstrate that MIM still does not scale very well, especially after a long enough (i.e., multi-stage) mask pre-training (i.e. MKD) procedure on the ImageNet-1K dataset. We still do not know how to improve the scalability, but perhaps this paper helps us exclude one candidate option. I would like to see the pinions of other reviewers for my final decision.

---

### Official Review · Reviewer_2u7d · 2023-11-01

**Soundness:** 3 good
**Presentation:** 4 excellent
**Contribution:** 3 good
**Rating:** 6
**Confidence:** 3

**Summary:**

This work studies the effect of knowledge distillation in mask auto-encoders. The authors observe that the choice of teacher representation becomes inconsequential when employing multi-stage distillation. As a result, the authors propose a novel approach called MKD, which utilizes bootstrapped teachers initialized randomly. Notably, MKD yields significant performance improvements when compared to alternative methods.

**Strengths:**

1. The paper presents a reasonable and novel story, and the conclusion regarding the teacher representation for Masked auto-encoders is convincingly demonstrated through solid preliminary experiments.

2. The organization and writing style are clear, making the paper easily readable.

3. The authors perform extensive and compelling experiments to validate the effectiveness of MKD, which results in significant performance improvements compared to the baselines.

4. The analysis provided in Section 6 is highly appreciated.

**Weaknesses:**

1.  The conclusion of the paper is limited to Masked auto-encoders where the teacher and student models are pre-trained with the same data (IN-1k). However, this is not clearly stated in the main paper.

2.  A While the authors mention the instability and sensitivity of other methods, it should be noted that MKD also requires careful pipeline design and hyper-parameter selection. For example,  MAE with fixed m (momentum updating teacher’s param) can achieve 84.3 at stage 2, however, in most cases in the ablation, MKD only outperforms MAE when the stage split number, epochs for each stage, and momentum parameter (m) and others are correctly set. This limits the practical application of MKD. This limits the practical application of MKD, as one could simply distill MAE for two stages with less time consumption and similar performance.

3. The paper only explores the fine-tune setting of semi-supervised learning (SSL), which diminishes the differences between pre-training models. How about the linear-probe setting? Would teachers also matter? Additionally, it is unclear whether the models mentioned in Section 6 are supervisedly fine-tuned on IN or pre-trained models before fine-tuning.

4. A minor weakness of the paper is that only l2 distillation is performed, neglecting other potential distillation methods.

**Questions:**

1. What is the recipe ⚗ in appendix A.1? I did not find the definition of this setting?

2. Pls see weakness.

---

> ### Author Response · Authors · 2023-11-13
>
> Thanks for your valuable comments and your appreciation for our technical contributions. We will clearly state the technical details in the revised version. Below, we discuss in detail the points you have raised.
>
> **Q1: Technical details.**
>
> In our pre-training and fine-tuning, we use IN-1K as the dataset and masked autoencoder as the student models. This is described in section 3 (common setup). We will clearly state the technical details in our revised version.
>
> **Q2: The hyper-parameter setting.**
>
> The setting for distilling MAE in the two-stage process is selected based on the ablation in section 5.4. In Table 1, all the results are set the same for a fair comparison. Distilling the mask autoencoder for two stages with any teacher achieves similar performance in classification, object detection, and segmentation.
>
> **Q3: Linear probing results and the details in section 6.**
>
> The linear probing results of dBOT using DINO as the teacher are 77.35%, while dBOT using other teachers achieves similar results (69.1%). We attribute this to the multi-crop technique and contrastive learning used in DINO. In section 6, all the models are pre-trained models before fine-tuning.
>
> **Q4: The meaning of recipe ⚗**
>
> The token ⚗ represents the distillation of dBOT with CLIP as the teacher, as described in appendix C.1. We will clearly state this in the revised version.
>
> We are investigating dBOT's effectiveness in other potential distillation methods except for l2 distillation. We appreciate your valuable feedback and will continue to explore and expand the scope of our work.

---

> > ### Comment · Reviewer_2u7d · 2023-11-23
> > **Thanks for the rebuttal.**
> >
> > Thank the authors for the rebuttal.
> > To some extent, I agree with the Reviewer aTJx that the main improvement of dBOT comes from multi-stage training. However, I still think the conclusion is valuable that selecting initial teacher representation is unnecessary for multi-stage training. This helps us exclude one candidate option to extend SSL training (as mentioned by Reviewer wVuK).
> > As a result, I still have a slightly positive perspective on this work, and I keep my score.

---

### Official Review · Reviewer_tvFW · 2023-11-01

**Soundness:** 3 good
**Presentation:** 4 excellent
**Contribution:** 3 good
**Rating:** 6
**Confidence:** 4

**Summary:**

The paper analyses the role of target representations (reconstruct targets) in masked knowledge distillation for self-supervised learning (SSL). It finds that distilling from the output of a randomly initialised network results in performance and properties similar to distilling from pre-trained representations. Based on these observations, the work proposes a new SSL method, dBOT that employs multiple stages of distillation starting from a randomly initialized teacher network. The results indicate that the proposed SSL pretraining consistently outperforms prior SSL pretraining on the downstream tasks of image classification, semantic segmentation, and object detection.

**Strengths:**

1. The paper makes a novel and interesting observation of the masked knowledge distillation being invariant to the initial teacher networks (Table 1).
2. The proposed method is simple and achieves consistent improvements over prior SSL methods (models) across multiple downstream tasks (Table 2,3,4).
3. The paper performs thorough ablation study (Table 5) and analysis of model weights and outputs (Section 6)
4. The method seems to be stable with respect to minor changes in the pretraining setup. (Table 5)
5. The work promises further improvements if data beyond ImageNet-1k is used (Appendix C.2 and C.3), including a result that shows model training from a CLIP-L teacher achieving 89.1% top-1 accuracy (new SoTA image recognition result)

**Weaknesses:**

Please look at the Questions section for suggestions on improving the draft.

Here are a few minor concerns (that are a bit related to each other):
1. **dBOT does not scale with more training?**. Results in Table 1 show a drop in performance on adding more stages, thus requiring a good stopping condition.
2. **Different models for different tasks** as opposed to prior works that typically release a single general model
The purpose of SSL is to learn a single model that generalize across tasks. Given that the performance on different downstream tasks peaks at different stages (Table 1), it is not clear how to choose the number of distillation stages to arrive at a single general model that can work across tasks (even beyond the ones discussed - eg. for embodied navigation [1]).
3. **Do the trends hold across random seeds?**. For a single random initialized teacher, Table 1 shows 2 stage being better for classification and stage 3 being better for detection/segmentation. It is not clear if trends like this (and the performance) hold across random seeds.

[1] Offline Visual Representation Learning for Embodied Navigation. Yadav et al. 2022.

**Questions:**

# Questions
1. In context of the weaknesses above, do authors have evidence that suggests that performance saturates and not drops with more stages and training?
2. **Sample efficiency.** SSL methods have been shown to improve with more training (Figure 7 of MAE [1]). Is the total number of pre-training iterations controlled across different methods in Table 2, 3, 4 and 5?
4. The authors compute properties **within** model and show that these match across different choices of teachers (Figure 2 and 3). Do the learned layers **across distilled models** show any correspondences? eg. how do the cross-model CKA similarity maps [2] look?
5. Table 6: Why is MAE faster than dBOT when both use an asymmetric encoder-decoder architecture?
6. "Additionally dBOT achieves more local attention than previous works". Can authors clarify this comment in context of Figure 2? It seems like MAE's attention distance plot is similar to the distilled models'.
7. Section 6. SVD line 5: Shouldn't lesser correlation in model's output result in lower redundancy?
8. C.1. I think I understand why the need for a decoder could be "eased when target contains high-level semantics". But can the authors elaborate a bit on why the "existence of decoder" may hurt?


# Minor typos/suggestions:
1. Section 1: A masked image **is** passed through the
2. Section 2.2: Page 3 last line: teacher and ~~bootstraps~~ bootstrap the teacher for stages
3. Table 2 caption: ~~Comparison~~ Comparing fine-tuning result
4. The method (section 4) suggests to continue repeating the distillation until a saturation in downstream task is observed. The authors may want to reword this as it is odd to have the SSL pretraining phase depend on a particular downstream task.

# References

[1] Masked Autoencoders Are Scalable Vision Learners. He et al. 2021

[2] Do Vision Transformers See Like Convolutional Neural Networks? Raghu et al. NeurIPS 2021

---

> ### Author Response · Authors · 2023-11-14
>
> Thanks for your valuable comments and your appreciation for our technical contributions. We will improve our draft following your suggestions. Below, we discuss in detail the points you have raised.
>
> **Q1: Does performance saturate and not drop with more stages and training?**
>
> We perform multi-stage training using pixels as the target. The fine-tuning performance on IN-1K at the 0th, 1st, 2nd, and 3rd stages is 83.6, 84.3, 84.4, and 84.3, respectively. The performance saturates after two-stage distillation and fluctuates around the saturation performance.
>
> **Q2: Different models for different tasks.**
>
> As we explained in response to Q1, the performance will fluctuate around the saturation performance. Downstream users can load any model after two-stage distillation with a slight performance difference.
>
> **Q3: Do the trends hold across random seeds?**
>
> During our experiments, we found that the performance of classification and detection is quite stable, except for the segmentation tasks. The segmentation performance reported in our paper is averaged over 5 random seeds, which we will clearly state in our revised version.
>
> **Q4: Is the total number of pre-training iterations controlled across different methods in Tables 2, 3, 4, and 5?**
>
> In Table 5, we conducted ablation studies to choose the best settings for dBOT. Then, the best setting was used in Tables 2, 3, and 4. The model performance saturates after multi-stage distillation.
>
> **Q5: Why is MAE faster than dBOT when both use an asymmetric encoder-decoder architecture?**
>
> MAE uses the original pixel as the target, which does not involve a teacher network, while dBOT uses an extra teacher to extract the reconstructed token.
>
> **Q6: It seems like MAE's attention distance plot is similar to the distilled models'.**
>
> They are similar but there are differences between them. Taking the last layer as an example, the maximum average attention distance for MAE is 85, while it is 70 for dBOT. The minimum average attention distance for MAE is 40, while it is 30 for dBOT. dBOT achieves more local attention than previous works.
>
> **Q7: Shouldn't a lesser correlation in the model's output result in lower redundancy?**
>
> We apologize for this mistake and will correct it in the revised version.
>
> **Q8: Why may the "existence of decoder" hurt?**
>
> The encoder and decoder are both used in the pre-training stage, but the decoder is discarded in the fine-tuning stage. When the target is low-level, the decoder is needed to decode the high-level representations encoded by the encoder. However, when the target contains high-level semantics, the existence of the decoder may even restrain the encoder from fully grasping knowledge from the teacher, leading to degraded performance.
>
> We appreciate your valuable feedback and will continue to explore and expand the scope of our work.

---

### Official Review · Reviewer_aTJx · 2023-11-01

**Soundness:** 2 fair
**Presentation:** 3 good
**Contribution:** 2 fair
**Rating:** 3
**Confidence:** 4

**Summary:**

This paper studies the role of target representation in Masked Image Modeling framework. Prior works each proposes a separate teacher network to generate the reconstruction target: BeiT uses DALL-E; MaskFeat uses HoG; MVP uses CLIP, without clearly justifying the necessity. This work finds that different choices of teacher network, including a random initialized one, leads to close performance in MIM training. Furthermore, it proposes a bootstrapped iterative MIM training pipeline, called dBOT, which shows improved performance.

**Strengths:**

The observation that diffrent teacher models do not make a large difference in generating target representation in MIM learning is interesting. The high-level idea of this paper is easy to follow, and the model does show good performance on classification, detection, and segmentation tasks.

**Weaknesses:**

1. The proposed method dBOT and the observation about the choice of teacher model is somewhat unrelated. It is my opinion that the good performance of dBOT comes from multi-stage training, which I find not clearly motivated in this paper. For instance, if the teacher network is switched in to some other pretrained networks, I feel confident that this dBOT pipeline would still perform good.

2. The observation of the insignificance of the choice of teacher network is somewhat aligned to the observation of MAE, which simply opts for raw image pixels and surpasses previous methods like BeiT or MaskFeat with more sophisticated target representation by a non-trivial margin. This is also validated from the results in appendix B.1, which shows *using the patch token obtained by a randomly initialized network as the target can achieve comparable results with a pixel as a target.*

3. The authors claim that *Using a random model as teachers not only avoids an extra pre-training stage, but also alleviates the painstaking selection of the target representations*. But I think this has already been achieved by MAE, since reconstrucing raw image pixels does not involve pretraining, and there is no need to select target representations. Thus, I am not sure what is the practical value of the observation of this work.

**Questions:**

1. In table 1, I find a randomly initialized teacher network could achieve 77.3 accuracy on ImageNet (and similar for other datasets), which really seems impossible to me. Am I missing something here?

---

> ### Author Response · Authors · 2023-11-13
>
> Thanks for your valuable comments. Below, we discuss in detail the points you have raised.
>
> **Q1: The relation between dBOT and the observation about the choice of teacher model.**
>
> Indeed, the performance of dBOT comes from the multi-stage training. The major finding in our paper is that the choice of the teacher model does not have to be carefully chosen if the distillation is done in stages. Specifically, we use a randomly initialized model as a teacher to formulate dBOT and compare the results with previous state-of-the-art (SOTA) models. dBOT still performs well when switching teacher networks since the choice of the teacher model is not crucial.
>
> **Q2: The observation is somewhat aligned with the observation of MAE.**
>
> In the early stages of Masked Image Modeling (MIM), researchers used different targets for performing MIM, such as pixels, Histogram of Oriented Gradients (HOG), and tokens from a pre-trained teacher. Researchers have different opinions on target selection and different results are achieved with different targets. MAE simplifies MIM by using pixels as the target and achieves better results than Beit and maskfeat, which demonstrates that using pixels is better than HOG and tokens from a pre-trained teacher. However, our paper finds that the choice of the teacher network is not crucial if the distillation is done in stages. In other words, similar results can be achieved by using any target (as shown in Table 1), even a randomly initialized teacher. This is the main difference between dBOT and MAE.
>
> **Q3: Reconstructing raw pixels.**
>
> MAE achieves good results by simply using pixels as a target. However, later works, like [1], achieve better results by using tokens from pre-trained teachers as targets. Different results are achieved with different targets. The motivation of our paper is that there is no work that provides a system-level study on the importance of how to choose an adequate target representation or teacher network to guide the learning of Masked Knowledge Distillation (MKD). Our work reveals that the choice of teacher is not crucial if the distillation is done in stages. In particular, we find that using a randomly initialized model as a teacher can achieve similar results through multi-stage distillation, which we formulate as dBOT. This alleviates the painstaking selection of target representations since we achieve similar performances with different target representations in multi-stage distillation.
>
> **Q4: The results of randomly initialized teacher.**
>
> The results of the 0th stage are the initialized teacher, and the results of the 1st and 2nd stages are the distilled student, which is described in Section 3.1. Therefore, 77.1 is the result of fine-tuning a randomly initialized model on IN-1K.
>
> [1] Wei, Yixuan, et al. "Contrastive learning rivals masked image modeling in fine-tuning via feature distillation." arXiv preprint arXiv:2205.14141 (2022).

---

> ### Author Response · Authors · 2023-11-14
> **highlight contributions**
>
> Here we would like to highlight the main contributions:
>
> a) **Pioneering System-Level Study in MIM with dBOT**: Our work, dBOT, stands as the first to conduct a system-level study on the role of the 'teacher' in Masked Image Modeling (MIM). While previous studies (such as MAE, MaskFeat, and MVP) suggest that the choice of target may not be pivotal for MIM's success, this hypothesis remained untested. dBOT is the first to empirically demonstrate this, marking the first contribution in the field.
>
> b) **Highlighting the Importance of Multi-Stage Masked Distillation**: We reveal that the key to effectively training high-capacity models lies in our novel multi-stage masked distillation pipeline, rather than in the meticulous selection of a training target. This approach contrasts with the typical one-stage training methodology prevalent in prior research, marking another contribution of this work.
>
> We are eager to engage in further discussions and questions!

---

> ### Comment · Reviewer_aTJx · 2023-11-19
> **Thanks for the clarification**
>
> Thanks for your prompt reply! I appreciate the potential effort it might take.
>
> However, after reading the authors' response, and opinions from other reviewers, I still feel this work has the following weakness:
>
> (1) As agreed by the authors, the superior performance of dBOT comes from its multi-stage training design, not from its analysis about the role of pre-trained teacher model. I feel not so comfortable that this paper tries to pitch its analysis "target representation" as the main driving force of its performance, as they are somewhat unrelated.
>
> (2) I agree that dBOT performs extensive analysis on the role of pre-trained teacher model in MIM, which is different from the raw pixel construction in MAE. However, I doubt if those analysis is really valuable, after MAE has proved that there is **no need** for such a pre-trained teacher. Indeed, as pointed out by Reviewer tvFW, dBOT is even slower than MAE due to its adoption of a teacher model. Plus, as I said earlier, there is *no painstaking selection of the target representations* if there is no teacher.
>
> (3) Following (2), one could argue that the fearetus extracted by a randomly initialized teacher is somewhat similar to raw pixels in the MIM framework, which is why those seemingly random features could work in the first place.
>
> (4) I checked [1] as the authors pointed out it *achieves better results by using tokens from pre-trained teachers as targets*. However, if my understanding is correct, [1] focuses on improving contrastive learning to the level of MAE by adopting stronger CLIP pre-trained teacher models. It is **not** a work of MIM. Hence, I doubt if that work could justify the motivation of this work, especially after there is already MAE. In other words, I cannot see any sign why we should adopt any teacher for targer representation in MIM. I feel confident that breaking MAE training into multiple stages would lead to just as good performance.
>
>
> My original review continues to apply.
>
>
> [1] Wei, Yixuan, et al. "Contrastive learning rivals masked image modeling in fine-tuning via feature distillation." arXiv preprint arXiv:2205.14141 (2022).

---

### Meta-Review · Area_Chair_TR83 · 2023-12-06

**Metareview:**

The paper proposes iterative "masked knowledge distillation" as a self-supervised feature learning approach. That is, masked modeling pre-training is performed initially using features from some feature extractor as targets, and then again iteratively using the features learned in the previous stage as targets. The paper shows that 1) this iterative process after ~2-3 steps converges to a similar representation quality independent of the initial feature targets, 2) this iterative process leads to some performance gains on some downstream tasks.

Even after rebuttal and discussion, the opinions of the reviewers are mixed. Below are some key pros and cons.

Pros:
1. Well written, easy to follow, fairly clear main idea
2. Simple method that shows performance gains over reasonable baselines
3. Thorough experiments, good ablation study and analysis

Cons:
1. The proposed method with multiple stage takes proportionally more compute to train - therefore the comparison to "single stage" baselines is not exactly fair
2. Somewhat limited performance gains, especially on classification
3. Potentially, the finding of the paper is a manifestation of limited scalability of masked modeling
4. Potentially, longer (and/or stage-wise) training of usual MAE would work similarly well, without any distillation

Overall, the paper seems novel and well executed and can be of interest for the readers. I therefore recommend acceptance. However, I would strongly encourage the authors to
(1) clearly highlight the computational costs of different methods
(2) add the results of MAE that is a) trained longer than usual (2x, 3x) , b) trained for several stages with learning rate schedule reset in between, but always RGB targets

**Justification For Why Not Higher Score:**

1. The proposed method with multiple stage takes proportionally more compute to train - therefore the comparison to "single stage" baselines is not exactly fair
2. Somewhat limited performance gains, especially on classification
3. Potentially, the finding of the paper is a manifestation of limited scalability of masked modeling
4. Potentially, longer (and/or stage-wise) training of usual MAE would work similarly well, without any distillation

**Justification For Why Not Lower Score:**

1.Well written, easy to follow, fairly clear main idea
2. Simple method that shows performance gains over reasonable baselines.
3. Thorough experiments, good ablation study and analysis

---

### Decision · Program_Chairs · 2024-01-16

Accept (poster)